# Service Quality in Tourism: A Systematic Literature Review and Keyword Network Analysis

**Jinsoo Park** [1]  **and EuiBeom Jeong** [2,*]

1   Department of Business Administration, The Catholic University of Korea, Jibong-ro, Bucheon-si, Gyeonggi-do 14662, Korea
2   Korea University Business School, Anam-ro, Seongbuk-gu, Seoul 02841, Korea
*   Correspondence: euibeom@korea.ac.kr; Tel.: +82-2-3290-2805

**Abstract:** The tourism industry has received increasing attention as it has become one of the fastest developing business sectors around the world. Service quality in tourism has come to be regarded as an important impetus for economic growth; however, the focus on tourism service quality has not yet been satisfactorily or comprehensively reviewed. Therefore, we conducted a systematic literature review combining bibliometric, citation network and keyword network analysis. We selected the top five tourism journals from the SCOPUS database and then collected papers according to their keywords. Our results revealed the critical issues, topics, and changes over time regarding research on tourism service quality. The critical issues included the important impact of service quality on tourist behavior and service quality evaluation, and topics comprised (1) tourist satisfaction, (2) sustainable issues in tourism, (3) value of service quality for customers, (4) restaurant service quality, (5) customers' perceptions of tourism, (6) service quality evaluation in tourism, and (7) service quality's influence on customer behavior. Furthermore, the keyword network analysis results revealed the most influential keywords.

**Keywords:** service quality; systematic literature review; keyword network analysis

## 1. Introduction

During the last few decades, the tourism industry has become extremely economically relevant, as it has become one of the fastest developing areas in the contemporary business environment. For example, global international tourist arrivals in 2013 reached a record of 1.087 billion, whereas international tourism revenue constituted a record US $1.159 billion in the same year [1]. This shows that tourism can significantly contribute to the economic growth of tourist destinations by increasing employment opportunities, developing infrastructure, and attracting foreign exchange earnings [2]. Tourism's economic impetus can be investigated using various perspectives, which include, for example, (1) its direct effects such as "sales, employment, tax revenues, and income levels", which come from the immediate impacts of tourist spending [2–4], (2) its indirect effects such as "prices, quantity and quality of products and services, taxes and property, and social and environmental impacts" [2–4], and (3) its efficiency and productivity due to economic resources being allocated to promote cost reductions in tourism sectors [2,5–7]. As a result, previous literature has regarded tourism as an important cause of variation in economic growth in many countries.

Among the factors related to tourism, service quality in tourism has received increasing academic attention. For example, research papers have been published in academic journals that utilize SERVQUAL, the most representative model for measuring service quality [8,9]. Nevertheless, service quality in tourism has not yet been satisfactorily reviewed despite its impact on tourists' destination choice. Furthermore, because of tourism's wide-ranging scope, which extends to various business

settings, few studies have used a comprehensive perspective to examine service quality in tourism. Therefore, our study's main objective was to identify the most influential studies as well as both broad and specific issues regarding tourism service quality and explore this research field's current and future directions and trends through a systematic literature review.

## 2. Literature Review

Identifying the themes of a given field for the purpose of improving our understanding of it, and therefore stimulating further research is a proactive effort. Through the process of mapping and evaluating the body of literature, scholars can identify potential research gaps and opportunities for future research [10].

With regards to conducting such literature reviews, researchers have adopted various approaches. One influential perspective is the "popularity-based approach," which includes the technique of bibliometric analysis. This method was created to investigate authors' keywords and titles of published articles in various research fields. Specifically, bibliometric analysis can provide insights not captured or evaluated by other reviews because it offers additional data regarding authors, affiliation, popular words, and keywords and their frequency of use. However, while the popularity-based approach can indicate the importance of titles or keywords in a given research area by investigating their frequency of use in papers, significant information can only be obtained post-publication [11]. Moreover, this approach is considered unsuitable for identifying shared topical content due to its inability to assess relationships among published papers within a certain field.

Another literature review method involves the "network-based approach," which uses citation and co-citation analysis to investigate the network structure that exists among articles in a given field by mapping and visualizing the citations generated among papers. Citation analysis has been used to determine the popularity of a publication [12]; that is, a network analysis of commonly used citations aims to identify the popularity of a published paper by counting how frequently a paper is cited by other papers. Unlike citation analysis, co-citation analysis has been implemented to identify topics in a given area by constructing a co-citation network comprising a set of nodes (journal papers) and a set of links (co-occurrences of the papers in other papers). That is, if two publications appear together in other publications' reference lists, they represent a co-citation relationship [13]. Co-citation can be used to explore data structure by applying mapping as a form of exploratory data analysis. In other words, a co-citation network constructed based on papers that are more frequently co-cited indicates that the included papers have similar subject areas [14]. However, those network-based analyses have mainly been conducted on published papers, rather than specified keywords. Therefore, a network of commonly used citations or co-citations does not directly represent a specified knowledge network for a given area from a comprehensive perspective. To comprehensively identify issues and topics pertaining to tourism service quality, it is therefore essential to include papers not involved in the network.

Lai et al. [15] performed a systematic literature review on service quality in Hospitality and Tourism using the pathfinder network approach. This study differs from the present study in two significant ways. First, Lai et al. used citation counts to identify the most influential papers for content analysis. While citation counts are an important indicator of a paper's impact, a highly cited paper can not necessarily indicate a prestigious paper, as measured by the number of times a paper is cited by other highly cited papers. In contrast, we used PageRank in the present study as an indicator of both "popularity, measured by citation count, and "prestige" to identify the most impactful papers and to analyze the contents of selected papers. Second, Lai et al. conducted a network-based study involving citation and co-citation analysis to identify research gaps and suggest directions for future research. Here, we adopted a systematic literature review approach that combined a network-based approach with bibliometric and keyword network analysis. The results are significantly different.

Considering these perceived drawbacks in the existing literature, we resolved to conduct a systematic literature review that combined the traditional systematic literature review approach

(bibliometric and citation analysis) with keyword network analysis for a more comprehensive evaluation of research on service quality in tourism. As part of our method, we identified and investigated the keyword network linking an author's keywords to comprehensively map knowledge of tourism service quality and identify important issues and topics and their change over time.

In our study, we first conducted rigorous bibliometric (i.e., frequency analysis) and network analysis on service quality in tourism research (e.g., citation, co-citation, and keyword network analysis) to map the knowledge structure of the issue and topic, and then carried out a content analysis of the key papers. In order to examine the structural characteristics of the network and identify critical issues and topics regarding service quality in tourism from a comprehensive perspective, we selected influential papers and extracted keywords. We performed a citation analysis to identify critical content in the existing literature and conducted a co-citation analysis to explore topical content based on classification of the existing studies. Finally, in order to carry out a systematic content analysis of the theme, we constructed a network based on keywords extracted from the selected publications and investigated changes in important keywords over time.

## 3. Research Methodology

As a first step, we selected tourism-related journals from the SCOPUS database. To extract the most influential papers, we reduced the journals to only the top five using the categories and impact factor (SJR, SCImago journal rank) provided by SCOPUS. We used "service quality" as the main search keyword.

We located papers with the identified keywords from the selected major journals by using the "article title, abstract, keywords" search in SCOPUS. To select the most influential papers, only journal articles written in English were used; conference papers, book series, commercial publications, and magazine articles were excluded. Search results included essential information such as title, author(s), abstract, paper keywords, and references.

Given that various tourism scholars might have differing perspectives in terms of which journals publish tourism-related articles, this study attempted to collect data from a diverse range of top tourism journals. The initial search attempts resulted in 178 papers. Due to a higher number of published papers in 2007 compared to 2008, we collected papers published during a 12-year period from 2008 to 2019. The results from the selected journals are summarized in Table 1. Figure 1 shows the quantity of publications per year.

**Table 1.** Selected journals (2008–2019).

| Journals | 2008 | 2009 | 2010 | 2011 | 2012 | 2013 | 2014 | 2015 | 2016 | 2017 | 2018 | 2019 | Total |
|---|---|---|---|---|---|---|---|---|---|---|---|---|---|
| Annals of Tourism Research | | | | | 1 | 1 | | | | | | | 2 |
| International Journal of Hospitality Management | 4 | 4 | 8 | 9 | 13 | 11 | 9 | 11 | 7 | 8 | 6 | 6 | 96 |
| Journal of Hospitality and Tourism Research | 2 | 1 | 6 | 1 | | 1 | 1 | 1 | 2 | 6 | 2 | | 23 |
| Journal of Travel Research | 1 | 4 | | | 1 | 1 | 1 | 1 | | 2 | 1 | 1 | 13 |
| Tourism Management | 3 | 2 | 1 | 4 | 3 | 1 | 3 | 5 | 6 | 4 | 10 | 2 | 44 |
| Total | 10 | 11 | 15 | 14 | 18 | 15 | 14 | 18 | 15 | 20 | 19 | 9 | 178 |

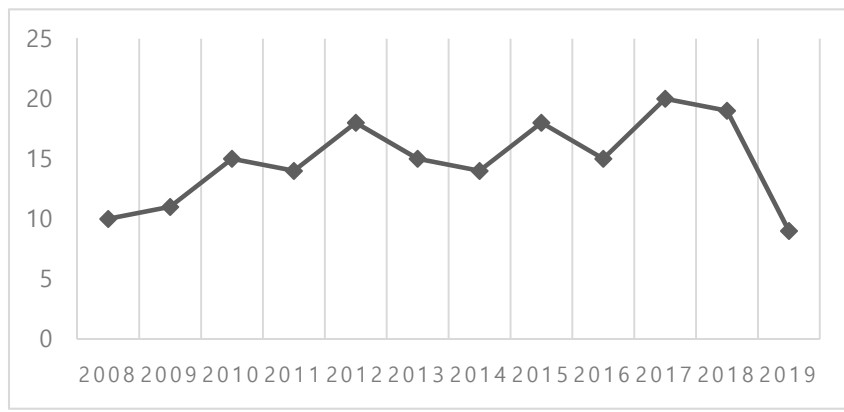

**Figure 1.** Quantity of publications per year (2008–2019).

Table 2 lists the journal, year, title, and author keywords associated with 178 research paper used in this research analysis. Using Excel, the data set was first arranged by journal name in ascending order. Then, the data set was arranged in ascending order by year and then title.

**Table 2.** Title and Author Keywords (2008–2019).

| No. | Journals | |
| --- | --- | --- |
| | Title (Year) | Author Keywords |
| | *Annals of Tourism Research* | |
| 1 | Tourist experience and Wetland parks: A case of Zhejiang, China (2012) | China; Post-trip behavioral intention; service quality; Tourist experience; Wetland parks |
| 2 | Value Co-creation significance of tourist resources (2013) | Effort; Involvement; Time spent; Tourist resources; Value perception |
| | *International Journal of Hospitality Management* | |
| 3 | Customer value in the hotel industry: What managers believe they deliver and what customer experience (2008) | Business hotels; Customer value; Hospitality industry; service quality |
| 4 | Determinants of dining satisfaction and post-dining behavioral intentions (2008) | Emotions; Loyalty; Recommendation; Restaurant services; Satisfaction; service quality |
| 5 | Does bureaucracy kill individual initiative? The impact of structure on organizational citizenship behavior in the hospitality industry (2008) | Bureaucracy; Centralization; Formalization; Hotel industry; Organizational citizenship behavior; service quality |
| 6 | The interaction of major resources and their influence on waiting times in a multi-stage restaurant (2008) | Capacity management; Restaurant resource; service quality; Simulation; Waiting time |
| 7 | Customers' cognitive, emotional, and actionable response to the servicescape: A test of the moderating effect of the restaurant type (2009) | Emotion; Perceived service quality; Revisit intention; servicescape; Theme restaurant |
| 8 | From public to private: Has employment practice changed in Bulgarian hotels? (2009) | Bulgaria; Emerging economies; Employment policy and practice; Hotels |
| 9 | Influence of institutional DINESERV on customer satisfaction, return intention, and word-of-mouth (2009) | Customer satisfaction; Food quality; Institutional DINESERV; Return intention; Word-of-mouth endorsement |
| 10 | Tipping behavior in Canadian restaurants (2009) | Canadian restaurant industry; CREST data; Gratuities; Tipping |
| 11 | An investigation of green hotel customers' decision formation: Developing an extended model of the theory of planned behavior (2010) | Decision-making; Extended theory of planned behavior; Green hotel; Revisit intention |
| 12 | Applying loss aversion to assess the effect of customers' asymmetric responses to service quality on post-dining behavioral intentions: An empirical survey in the restaurant sector (2010) | Customers' post-behavioral intentions; DINESERV; Loss aversion |
| 13 | Effects of organizational/occupational characteristics and personality traits on hotel manager emotional exhaustion (2010) | Burnout; Emotional exhaustion; Hotel; Personality traits |
| 14 | Effects of service quality and food quality: The moderating role of atmospherics in an ethnic restaurant segment (2010) | Atmospherics; Ethnic restaurant; Food quality; Korean restaurant; Loyalty; Moderating effect; Satisfaction; service quality |
| 15 | Exploratory assessment of the Macao casino dealers' job perceptions (2010) | Casino dealers; Front-line employees; Job perceptions; Macao |

**Table 2.** *Cont.*

| No. | Journals | |
| --- | --- | --- |
| | **Title (Year)** | **Author Keywords** |
| | International Journal of Hospitality Management | |
| 16 | Lifestyle businesses: Insights into Blackpool's hotel sector (2010) | Blackpool hotels; Lifestyle entrepreneurs; Micro-businesses |
| 17 | Pay for performance and work attitudes: The mediating role of employee-organization service value congruence (2010) | P-O fit; Pay for performance; service value congruence; Work attitudes |
| 18 | Supply management practices and performance in the Canadian hospitality industry (2010) | Hospitality; Performance; Purchasing strategy; service; Supply management |
| 19 | An examination of electronic video clips in the context of hotel Websites (2011) | E-VISQUAL; Experiential services; Video clip; Virtual human interaction; Visual image |
| 20 | Customers' perceptions of service quality: Do servers' age stereotypes matter? (2011) | Age stereotypes; Perceptions of service quality; Server |
| 21 | Foreign travelers' satisfaction with traditional Korean restaurants (2011) | Customer satisfaction; Expectation; Korean restaurants; service quality; Value for money |
| 22 | Luxury marketing: The influences of psychological and demographic characteristics on attitudes toward luxury restaurants (2011) | Demographics; Hedonism; Luxury marketing; Materialism; Perfectionism; Uniqueness |
| 23 | Restaurant consumers repeat patronage: A service quality concern (2011) | Cleanliness; Consumers; Repeat patronage; Restaurant; service quality |
| 24 | Restaurant experiences triggering positive electronic word-of-mouth (eWOM) motivations (2011) | Electronic word-of-mouth (eWOM); EWOM motivations; Perceived service quality; Restaurant experiences |
| 25 | Reward climate and its impact on service quality orientation and employee attitudes (2011) | Job satisfaction; Organizational commitment; Reward climate; service quality orientation (SQO) |
| 26 | Strategic alignment leverage between hotels and companies: The buyer-supplier relationship perspective (2011) | Environment uncertainty; service vision; Strategic alignment |
| 27 | When will customers care about service failures that happened to strangers? The role of personal similarity and regulatory focus and its implication on service evaluation (2011) | Attribution; Personal similarity; Regulatory focus; service failures |
| 28 | Customer opportunistic complaints management: A critical incident approach (2012) | Complaining behavior; Critical incident technique; Opportunistic complaint; Unethical consumer behavior |
| 29 | Development and validation of the casino service quality scale: CASERV (2012) | Casino; Macau; Scale development; service quality |
| 30 | Does better service induce higher profitability? Evidence from Taiwanese Hospitality Industry (2012) | International tourist hotel; Profitability; service quality |
| 31 | Enhancing service quality improvement strategies of fine-dining restaurants: New insights from integrating a two-phase decision-making model of IPGA and DEMATEL analysis (2012) | DEMATEL; IPGA; Restaurant service quality |
| 32 | Factorial validation of hospitality service attitude (2012) | Customer satisfaction; Hotel industry; Interaction quality; service attitude; service quality |
| 33 | It is all about the emotional state: Managing tourists' experiences (2012) | Customer satisfaction; Emotional state; Hotel setting; Leisure experience; servicescape |
| 34 | Measurement scale for eco-component of hotel service quality (2012) | Eco-component; Eco-label; Ecological expectations; Hotel service quality; Wellness hotels |
| 35 | Relationship quality between exhibitors and organizers: A perspective from Mainland China's exhibition industry (2012) | China; Exhibitions; Relationship quality |
| 36 | service guarantees in the hotel industry: Their effects on consumer risk and service quality perceptions (2012) | Corporate reputation; Hotel; Perceived quality; Perceived risk; service guarantees |
| 37 | service quality and tipping: The moderating role of the quality of food (2012) | Quality of food; service quality; Tipping behavior |
| 38 | service-leadership competencies for hospitality and tourism management (2012) | Competency model; Hospitality and tourism; Leadership; service quality |
| 39 | The underlying dimensions of tipping behavior: An exploration, confirmation, and predictive model (2012) | Consumer behavior; Motivation; Restaurants; service quality; Social norms; Tipping |
| 40 | When I put on my service mask: Determinants and outcomes of emotional labor among hotel service providers according to affective event theory (2012) | Affective event theory; Emotional labor; Hotel service employees; Negative emotions; service quality |
| 41 | Corporate social responsibility practices in four and five-star hotels: Perspectives from Hong Kong visitors (2013) | Corporate social responsibility; Customer perceptions; Hong Kong; Hotels |

**Table 2.** *Cont.*

| No. | Journals | |
| --- | --- | --- |
| | **Title (Year)** | **Author Keywords** |
| | *International Journal of Hospitality Management* | |
| 42 | Emotional intelligence and adaptability-service encounters between casino hosts and premium players (2013) | Adaptability; Casino hosts; Casino industry; Emotional intelligence; Premium player; service performance |
| 43 | Factors influencing internal service quality at international tourist hotels (2013) | Hospitality service; Internal service quality; Leadership; Organizational culture |
| 44 | Factors influencing the effectiveness of online group buying in the restaurant industry (2013) | Discount; Group buying; Restaurant; Return intention; Satisfaction; service quality |
| 45 | Is satisfaction enough to ensure reciprocity with upscale restaurants? The role of gratitude relative to satisfaction (2013) | Gratitude; Reciprocal behaviors; Relationship benefits; Satisfaction; Upscale restaurant |
| 46 | Justice perceptions and drives of hotel employee social loafing behavior (2013) | China hotel industry; Commitment; Justice perceptions; Social loafing; Turnover intention |
| 47 | Multi-dimensions of patrons' emotional experiences in upscale restaurants and their role in loyalty formation: Emotion scale improvement (2013) | Commitment; Emotion measurement; Image; Loyalty intentions; Satisfaction; service quality; Trust; Upscale restaurant |
| 48 | The effect of regulatory focus and delay type on consumers' reactions to delay (2013) | Affective response; Delay type; Expectancy model; Field theory; Regulatory focus; service quality |
| 49 | The effects of restaurant servers' perceptions of customers' tipping behaviors on service discrimination (2013) | Buyer monitoring; Employee control mechanisms; Restaurant; service discrimination; service quality; Tipping |
| 50 | The influence of customer brand identification on hotel brand evaluation and loyalty development (2013) | Brand loyalty; Brand trust; Identification; Perceived value; service quality |
| 51 | The role of frequency of patronage and service quality of all-you-can-eat buffet restaurant: A perspective of socio-economic and demographic characteristics of African American consumers (2013) | African American consumers; Blacks; Buffet restaurant marketing; Food consumption; Restaurant patronage; service quality |
| 52 | A patron, a referral and why in Macau casinos -The case of mainland Chinese gamblers (2014) | Attitudinal loyalty; Behavioural loyalty; Casino service quality; Chinese gamblers; Loyalty programme; Macau casinos; Membership |
| 53 | Applying service Profit Chain model to the Korean restaurant industry (2014) | Customer constructs; Employee constructs; Internal service quality; Korean restaurant; Organizational commitment; service Profit Chain |
| 54 | Attitudinal dimensions of professionalism and service quality efficacy of frontline employees in hotels (2014) | Customer orientation; Knowledge; Professionalism; Self-efficacy; Self-management; Sense of calling |
| 55 | Becoming an ambidextrous hotel: The role of customer orientation (2014) | Customer orientation; Exploitation; Exploration; service improvement; service innovation |
| 56 | Effective restaurant rating scale development and a mystery shopper evaluation approach (2014) | Mystery shopper; Restaurant; Restaurant rating; Scale development; service quality |
| 57 | Job standardization and service quality: The mediating role of prosocial service behaviors (2014) | Job standardization; Prosocial service behaviors; service quality |
| 58 | Linking service quality, customer satisfaction and loyalty in casinos, does membership matter? (2014) | Casino marketing, Macau; Customer loyalty; Customer satisfaction; Membership; service quality |
| 59 | Why do customers switch? More satiated or less satisfied (2014) | Atmospheric quality; Food quality; Satiation; Satisfaction; service quality; Switching intention |
| 60 | Wine attributes, perceived risk and online wine repurchase intention: The cross-level interaction effects of website quality (2014) | Hierarchical linear modeling; Multilevel analysis; Online wine repurchase intention; Perceived risk; Website quality; Wine attributes |
| 61 | A conflict of choice: How consumers choose where to go for dinner (2015) | Food quality; Multi-attribute decision making; Restaurant choice experiment; service quality |
| 62 | A segmentation of online reviews by language groups: How English and non-English speakers rate hotels differently (2015) | Language; Online ratings; Reputation; Satisfaction difference; Traveler distribution |
| 63 | Antecedents of mandatory customer participation in service encounters: An empirical study (2015) | Customer participation; Loyalty; Purchase importance; Role clarity; Self-efficacy; service quality |
| 64 | Applying mixed methods to identify what drives quick service restaurant's customer satisfaction at the unit-level (2015) | Customer satisfaction; Kano's model; Performance optimization; Product quality; Quick service restaurants; service quality |
| 65 | Consumer-based chain restaurant brand equity, brand reputation, and brand trust (2015) | Brand equity; Brand reputation; Brand trust; Chain restaurant; Visit purpose |
| 66 | Does uncertain demand affect service quality? (2015) | Capital-labor ratio; Hotel industry; service quality; Uncertain demand |
| 67 | Impacts of cruise service quality and price on vacationers' cruise experience: Moderating role of price sensitivity (2015) | Cruise; Loyalty; Novelty; Price; Price sensitivity; service quality; Value |

**Table 2.** *Cont.*

| No. | Journals | |
| --- | --- | --- |
| | **Title (Year)** | **Author Keywords** |
| | *International Journal of Hospitality Management* | |
| 68 | Positive emotions and behavioral intentions of customers in full-service restaurants: Does aesthetic labor matter? (2015) | Aesthetic labor; Behavioral intention; Full-service restaurant; Positive emotion |
| 69 | The cross-impact of network externalities on relationship quality in exhibition sector (2015) | Customer loyalty; Exhibition; Network externalities; Relationship quality; service quality |
| 70 | Understanding service experience and its impact on brand image in hospitality sector (2015) | Brand image; Customer experiences; Customer Satisfaction; India; service quality |
| 71 | Value dimensions in consumers' experience: Combining the intra- and inter-variable approaches in the hospitality sector (2015) | Extrinsic vs. intrinsic values; Loyalty; Perceived value; Satisfaction; Value dimensions |
| 72 | Adaptation of hospitality service quality scales for marina services (2016) | Marinas; service quality; Turkish marinas; Yachtsmen's satisfaction |
| 73 | An emotional labor perspective on the relationship between customer orientation and job satisfaction (2016) | Customer orientation; Emotional labor; Job satisfaction |
| 74 | Do competitive strategy effects vary across hotel industry cycles? (2016) | Advertising expenditure; Business cycle; Hotel industry; Pricing strategy; Quality strategy |
| 75 | Embracing or fighting the urge: A multilevel investigation on casino service, branding and impulsive gambling (2016) | Brand attachment; Brand equity; Casinos; Impulsive gambling; Loyalty; service quality |
| 76 | Hedonic adaptation and satiation: Understanding switching behavior in the restaurant industry (2016) | Hedonic adaptation; Restaurants; Satiation; Satisfaction; Switching |
| 77 | Internal branding process: Exploring the role of mediators in top management's leadership-commitment relationship (2016) | Brand commitment; Employee brand knowledge; Employee-brand fit; Psychological contract; Top management's brand-oriented leadership |
| 78 | The influences of restaurant menu font style, background color, and physical weight on consumers' perceptions (2016) | Menu background color; Menu font style; Menu weight; Restaurant scale; Restaurant service |
| 79 | Exploring the nonlinear impact of critical incidents on customers? general evaluation of hospitality services (2017) | Critical incident technique; Hospitality industry; Kano model; service quality; TripAdvisor |
| 80 | Extending the scope of hotel client reactions to employee injustice: Hotel employer reviews on the Internet (2017) | Client satisfaction; Hotel employee (mis)treatment; Hotel management; Organizational justice; The Internet |
| 81 | Impact of hotel-restaurant image and quality of physical-environment, service, and food on satisfaction and intention (2017) | Customer satisfaction; Food quality; Intentions; Luxury hotel restaurant; Physical environment quality; service quality |
| 82 | The effects of teppanyaki restaurant stimuli on diners? Emotions and loyalty (2017) | Chef's image; Diners; Emotions; Loyalty; Teppanyaki |
| 83 | The impact of personal and functional aspects of restaurant employee service behaviour on customer satisfaction (2017) | Customer satisfaction; Personal aspects; Restaurant service quality; service behaviour; service encounter |
| 84 | The impact of the social servicescape, density, and restaurant type on perceptions of interpersonal service quality (2017) | Density; DINESERV; Restaurant; service quality; Social servicescape |
| 85 | Understanding the dimensions of customer relationships in the hotel and restaurant industries (2017) | Customer lifetime financial value; Customer relationships; Hospitality marketing strategy; Relationship marketing |
| 86 | Willingness to pay in negative restaurant service encounters (2017) | Ambiance; Food quality; service encounter; service quality; Value; Willingness to pay |
| 87 | Projecting service quality: The effects of social media reviews on service perception (2018) | Social media reviews; Social media popularity; Service quality; Credibility; Service attributes; Facebook |
| 88 | Less is more: A new insight for measuring service quality of green hotels (2018) | and GLSERV scale; Carbon reduction; Environmental protection; Green hotel; service quality |
| 89 | On the relationship between intellectual capital and financial performance: A panel data analysis on SME hotels (2018) | Dynamic panel data; Financial performance; Intellectual capital; Small and medium-sized enterprise hotels |
| 90 | Projecting service quality: The effects of social media reviews on service perception (2018) | Credibility; Facebook; service attributes; service quality; Social media popularity; Social media reviews |
| 91 | The effects of organizational and personal resources on stress, engagement, and job outcomes (2018) | Customer orientation; Engagement; Hotel employees; Job outcomes; Job stress; Management commitment to service quality |
| 92 | The Integration between service Value and service Recovery in the Hospitality Industry: An Application of QFD and ANP (2018) | Analytic network process; Customer satisfaction; Quality function deployment; service failure; service recovery |

**Table 2.** *Cont.*

| No. | Journals | |
|---|---|---|
| | **Title (Year)** | **Author Keywords** |
| | *International Journal of Hospitality Management* | |
| 93 | Changing tires on a moving car: the role of timing in hospitality and service turnaround processes (2019) | Hospitality industry; Recovery; Retrenchment; service industry; Turnaround process |
| 94 | Consumer values and service quality perceptions of food truck experiences (2019) | DINESERV; Food truck; Hedonic value; Importance-performance analysis; Utilitarian value |
| 95 | Developing and validating a mobile catering app success model (2019) | E-commerce system success model; eWOM; Mobile catering app; Perceived price; Perceived promotions; Product quality |
| 96 | Exploring Airbnb service quality attributes and their asymmetric effects on customer satisfaction (2019) | Airbnb; Customer satisfaction; Impact asymmetry analysis; Impact range performance analysis; Mixed method; service quality |
| 97 | Good discounts earn good reviews in return? Effects of price promotion on online restaurant reviews (2019) | Image; Online consumer review; Price promotion; Restaurant; Review rating; Textual content |
| 98 | The impact of green practices in coastal tourism: An empirical investigation on an eco-labelled beach club (2019) | Beach club; Behavioral intention; Coastal tourism; Ecolabel; Green practices; PLS-SEM |
| | *Journal of Hospitality and Tourism Research* | |
| 99 | Relationships Among Experiential Marketing, Experiential Value, and Customer Satisfaction (2008) | customer satisfaction; experiential marketing; experiential value; structural equation modeling |
| 100 | Tourists' Satisfaction Judgments: An Investigation of Emotion, Equity, and Attribution (2008) | attribution; culture; emotion; equity; package tour; satisfaction |
| 101 | Stakeholder service perspectives: A triadic analysis of service quality in south Mississippi fine dining restaurants (2009) | Quality perceptions; service quality; Stakeholders |
| 102 | Does cultural difference affect customer's response in a crowded restaurant environment? A comparison of American versus Chinese customers (2010) | Attribution; Crowdedness; Cultural differences |
| 103 | Influence of the Quality of Food, service, and Physical Environment on Customer Satisfaction and Behavioral Intention in Quick-Casual Restaurants: Moderating Role of Perceived Price (2010) | behavioral intention; perceived price; quality dimensions (food service and physical environment); quick-casual restaurants; satisfaction |
| 104 | The role and effect of job satisfaction and empowerment on customers' perception of service quality: A study in the restaurant industry (2010) | Customer satisfaction; Employee empowerment; Employee job satisfaction; HRM-service quality link; Restaurant services management; service quality |
| 105 | Tipping and service quality: A within-subjects analysis (2010) | Equity; Incentive; Reward; service quality; Tipping |
| 106 | Tour guide performance and tourist satisfaction: A study of the package tours in Shanghai (2010) | service quality; Tour experience; Tour guide performance; Tour operator; Tourist satisfaction |
| 107 | Toward a Theory of Restaurant Décor: An Empirical Examination of Italian Restaurants in Manhattan | New York restaurants; restaurant décor; theory of reasoned action |
| 108 | Synthesizing the effects of service quality, value, and customer satisfaction on behavioral intentions in the motel industry: An empirical analysis (2011) | behavioral intentions; customer satisfaction; motel industry; service quality; value |
| 109 | An Exploratory Examination of Supervisor Undermining, Employee Involvement Climate, and the Effects on Customer Perceptions of service quality in Quick-service Restaurants (2013) | customer service; employee involvement; quick-service restaurants; undermining |
| 110 | The Influence of Hotel Price on Perceived service quality and Value in E-Tourism: An Empirical Investigation Based on Online Traveler Reviews (2014) | online reviews; perceived quality; perceived value; price effects |
| 111 | Enhancing Consumer Value in Wine Tourism (2015) | consumption values; experiential marketing; marketing; service quality |
| 112 | A quality-Value-Attitude Model: The Case of Expo 2010 Shanghai (2016) | attitude; economic value; emotional value; mega-event quality; Shanghai Expo |
| 113 | Socially Responsible Customers and the Evaluation of service quality (2016) | customer expectations; dimensions of social responsibility; ethical consumers; service quality; social responsibility dimension scale; socially responsible consumers |
| 114 | Construction and Validation of the Customer Participation Scale (2017) | customer loyalty; customer roles; mandatory customer participation; scale development; service quality |
| 115 | Signaling service quality via Website e-CRM Features: More Gains for Smaller and Lesser Known Hotels | Signaling theory; electronic customer relationship management; e-CRM, website quality |

**Table 2.** *Cont.*

| No. | Journals | |
|---|---|---|
| | **Title (Year)** | **Author Keywords** |
| | *Journal of Hospitality and Tourism Research* | |
| 116 | Examination of Restaurant Quality, Relationship Benefits, and Customer Reciprocity from the Perspective of Relationship Marketing Investments (2017) | customer reciprocity; relationship benefit; relationship marketing investment; restaurant quality |
| 117 | Psychological Capital in the Quick Service Restaurant Industry: A Study of Unit-Level Performance (2017) | customer satisfaction; psychological capital; quick service restaurants; revenues; service quality |
| 118 | Signaling service quality via Website e-CRM Features: More Gains for Smaller and Lesser Known Hotels (2017) | e-CRM; electronic customer relationship management; signaling theory; website quality |
| 119 | The Effect of Tourist Relationship Perception on Destination Loyalty at a World Heritage Site in China: The Mediating Role of Overall Destination Satisfaction and Trust (2017) | destination image; loyalty; satisfaction; service fairness; service quality; trust |
| 120 | The Effect of Event Supportive Service Environment and Authenticity in the Quality-Value-Satisfaction Framework (2018) | authenticity; customer satisfaction; festival; perceived value; program quality; service environment |
| 121 | Tourist Shoppers??Evaluation of Retail service: A Study of Cross-Border Versus International Outshoppers (2018) | cross-border outshoppers; evaluation of retail service; international outshoppers; tourist shopping behavior |
| | *Journal of Travel Research* | |
| 122 | The effects of quality and satisfaction on awareness and behavioral intentions: Exploring the role of a wine festival (2008) | Awareness; Behavioral intentions; Perceived quality; Satisfaction; Wine festival |
| 123 | "You felt like lingering": Experiencing "real" service at the winery tasting room (2009) | Hospitality; service experience; service quality; Wine tourism |
| 124 | A mediation model of tourists' repurchase intentions for packaged tour services (2009) | Packaged tour services; Perceived value; Repurchase intentions; Satisfaction; service quality |
| 125 | Small-business owners' knowledge and rural tourism establishment performance in Spain (2009) | Knowledge; Objective quality; Perceived quality; Rural tourism; Small-business owner |
| 126 | Tip-Collection strategies, service guarantees, and consumer evaluations of group package tours (2009) | Consumer evaluations; Group package tours; service guarantee; Tip-collection strategy |
| 127 | The role of cybermediaries in reputation building and price premiums in the online hotel market (2012) | cybermediary; online hotel market; online travel agent; reputation |
| 128 | A Systematic Approach to Scale Development in Tourist Shopping Satisfaction: Linking Destination Attributes and Shopping Experience (2013) | experience; satisfaction; scale development; tourism shopping; tourist facility |
| 129 | Guests and Hosts Revisited: Prejudicial Attitudes of Guests toward the Host Population (2014) | cultural bubble; host-guest relationships; prejudice; tourism's impact; tourist attitudes |
| 130 | Perceived Destination Image: An Image Model for a Winter Sports Destination and Its Effect on Intention to Revisit (2015) | destination image model; SEM; service quality; sport events; visitor management |
| 131 | Enhancing service Loyalty: The Roles of Delight, Satisfaction, and service quality (2017) | delight; experience; loyalty; PLS path modeling; quality; satisfaction |
| 132 | Service Quality Perceptions, Online Visibility, and Business Performance in Rural Lodging Establishments (2017) | business performance; online comments; qualitative content analysis; rural tourism; web visibility |
| 133 | Medical Tourism Experience: Conceptualization, Scale Development, and Validation (2018) | health tourism; tourist perception; travel experience; wellness tourism |
| 134 | Flying to Quality: Cultural Influences on Online Reviews (2019) | airlines; cultural differences; electronic WOM; online reviews; service quality; structural topic model |
| | *Tourism Management* | |
| 135 | A service quality measurement architecture for hot spring hotels in Taiwan (2008) | Analysis network process (ANP); Hot spring hotels; service quality |
| 136 | Developing a multidimensional and hierarchical service quality model for the travel agency industry (2008) | Multidimensional and hierarchical structure; service quality; Travel agencies industry |
| 137 | Do airline self-service check-in kiosks meet the needs of passengers? (2008) | Importance-performance analysis; Ridit analysis; Self-service kiosks |
| 138 | A multi-criteria assessment of tourist farm service quality (2009) | DEXi; Multi-criteria modeling; Rural tourism; service quality; Tourist farm |
| 139 | Understanding the relationships of quality, value, equity, satisfaction, and behavioral intentions among golf travelers (2009) | Behavioral intention; Customer loyalty; Equity; Golf traveler; Satisfaction; Value |
| 140 | A visitors' evaluation index for a visit to an archaeological site (2010) | Archaeological site; Formative index; service convenience; service experience; service quality |
| 141 | An exploratory inquiry into destination risk perceptions and risk reduction strategies of first time vs. repeat visitors to a highly volatile destination (2011) | Motives; Repeat visitors; Risk perception; Risk reduction strategies; Volatile destination |

**Table 2.** *Cont.*

| No. | Journals | |
| --- | --- | --- |
| | **Title (Year)** | **Author Keywords** |
| | Tourism Management | |
| 142 | Critical issues affecting the service quality and professionalism of the tour guides in Hong Kong and Macau (2011) | Entrepreneur role; Professional habitus; Professionalism; Role conflict; service quality; Tour guide |
| 143 | Customer satisfaction using low cost carriers (2011) | Behavioral intentions; Customer satisfaction; Low cost carriers; Perceived service quality |
| 144 | Using a modified grey relation method for improving airline service quality (2011) | Airline; Customers' needs; Grey relation; Multiple-criteria decision-making (MCDM); service quality; SERVQUAL |
| 145 | Comment on "Using a modified grey relation method for improving airline service quality" (2012) | Airline; Grey relation; MCDM; service quality; VIKOR |
| 146 | Passengers' perceptions of airline lounges: Importance of attributes that determine usage and service quality measurement (2012) | Airline lounge; Atmosphere; F&B service; service quality |
| 147 | Reply to "Comment on using a modified grey relation method for improving airline service quality" (2012) | Grey relation; service quality; TOPSIS; VIKOR |
| 148 | Quality deterioration in package tours: The interplay of asymmetric information and reputation (2013) | Asymmetric information; China; Package tours; Quality deterioration; Reputation |
| 149 | A novel framework for customer-driven service strategies: A case study of a restaurant chain (2014) | Customer satisfaction; Importance-performance analysis; service quality; Signal-to-noise ratio; The Kano model |
| 150 | Examining strategies for maximizing and utilizing brand prestige in the luxury cruise industry (2014) | Brand consciousness; Brand identification; Brand prestige; Luxury cruise; Well-being perception |
| 151 | Improving importance-performance analysis: The role of the zone of tolerance and competitor performance. The case of Taiwan's hot spring hotels (2014) | Benchmarking; Hot spring hotels; services quality; Zone of tolerance |
| 152 | Developing similarity-based IPA under intuitionistic fuzzy sets to assess leisure bikeways (2015) | Bikeway; Importance-performance analysis; Intuitionistic fuzzy set; Pattern recognition; service quality; Similarity |
| 153 | Ensuring corporate travel compliance-Control vs. commitment strategies (2015) | Case study; Commitment; Control; Corporate travel; service quality; service triad; Travel policy compliance |
| 154 | service quality and the training of employees: The mediating role of organizational commitment (2015) | Commitment; Hotels; India; service quality; Tourist; Training |
| 155 | The determinants of recommendations to use augmented reality technologies: The case of a Korean theme park (2015) | Augmented reality; DeLone and McLean model; Personal innovativeness; Process theory; Satisfaction; Smartphone |
| 156 | Using a randomised experiment to test the causal effect of service quality on visitor satisfaction and loyalty in a remote national park (2015) | Interventions; Loyalty; Protected areas; Randomised experiment; Visitor satisfaction |
| 157 | A comparison of service quality attributes for stand-alone and resort-based luxury hotels in Macau: 3-Dimensional importance-performance analysis (2016) | Importance-performance analysis; Luxury hotel; service quality measurement scale; Three-factor theory |
| 158 | Hotel attributes: Asymmetries in guest payments and gains—A stated preference approach (2016) | Discrete choice experiments; Discrete choice models; Hotel choice; Willingness to accept; Willingness to pay |
| 159 | Police culture, tourists and destinations: A study of Uttarakhand, India (2016) | Leader behavior; Police organization culture; service quality of police; Tourists' confidence in police |
| 160 | The effects of perceived service quality on repurchase intentions and subjective well-being of Chinese tourists: The mediating role of relationship quality (2016) | Customer satisfaction; Customer-company identification; Repurchase intentions; service quality; Subjective well-being |
| 161 | The trickle-down effect of servant leadership on frontline employee service behaviors and performance: A multilevel study of Chinese hotels (2016) | Hospitality; Servant leadership; service climate; service quality; service-oriented behavior; Trickle-down model |
| 162 | Travel web-site design: Information task-fit, service quality and purchase intention (2016) | Empirical research; Information-task fit; Product quality; Purchase intentions; Website design quality |
| 163 | An ant colony based optimization for RFID reader deployment in theme parks under service level consideration (2017) | Ant colony optimization; Reader deployment; service level index; Theme parks; Tracking systems |
| 164 | Assessing the effectiveness of empowerment on service quality: A multi-level study of Chinese tourism firms (2017) | Cross-level; Empowerment climate; Psychological empowerment; service behavior-based evaluation (SBE); service quality; Tourism firms |
| 165 | Festival attributes and perceptions: A meta-analysis of relationships with satisfaction and loyalty (2017) | Attributes; Festival; Loyalty; meta-analysis; Perceptions; Satisfaction |

**Table 2.** *Cont.*

| No. | Journals | |
|---|---|---|
| | **Title (Year)** | **Author Keywords** |
| | Tourism Management | |
| 166 | Sources of satisfaction with luxury hotels for new, repeat, and frequent travelers: A PLS impact-asymmetry analysis (2017) | Frequent travelers; Impact-asymmetry analysis; Luxury hotels; Satisfaction; service quality |
| 167 | Contemplating museums??service failure: Extracting the service quality dimensions of museums from negative on-line reviews (2018) | Museum management; Museum tourism; On-line review; service failure; service quality; Social media; TripAdvisor; Visitor experience |
| 168 | Does a happy destination bring you happiness? Evidence from Swiss inbound tourism (2018) | Happiness; Life satisfaction; Switzerland; Tourist destination |
| 169 | In-flight NCCI management by combining the Kano model with the service blueprint: A comparison of frequent and infrequent flyers (2018) | Airline industry; Flying frequency; In-flight NCCI; Kano model; service blueprint |
| 170 | Innovation and 19th century hotel industry evolution (2018) | Hotel industry history; Niche cumulation; Technological transition; Tourism history; Tourism methodology |
| 171 | Is culture of origin associated with more expressions? An analysis of Yelp reviews on Japanese restaurants (2018) | Cross-cultural difference; Online restaurant review; Sentiment analysis; Vocabulary range |
| 172 | Is role stress always harmful? Differentiating role overload and role ambiguity in the challenge-hindrance stressors framework (2018) | Challenge-hindrance stressors; Employee psychological empowerment; Hierarchical linear modeling; Organizational supportive leadership climate; Role ambiguity; Role overload; service quality |
| 173 | Mobile social tourism shopping: A dual-stage analysis of a multi-mediation model (2018) | Artificial Neural Network analysis; Mobile social tourism shopping; Multiple mediation analysis; Partial Least Squares Structural Equation Modelling; Stimulus-Organism-Response framework; Tourism products and services |
| 174 | Past themes and future trends in medical tourism research: A co-word analysis (2018) | Bibliometric analysis; Co-word analysis; Health tourism; Medical tourism; Thematic evolution |
| 175 | Predicting determinants of hotel success and development using Structural Equation Modelling (SEM)-ANFIS method (2018) | Critical Success Factors (CSFs); HOT-fit Model; Hotel success and development; SEM-ANFIS; TOE framework; Tourism |
| 176 | Quality assessment of airline baggage handling systems using SERVQUAL and BWM (2018) | Baggage handling; Best worst method; BWM; Quality; SERVQUAL |
| 177 | Cooperation and competition between online travel agencies and hotels (2019) | competition; Cooperation; hotel; O2O commerce; Online travel agency |
| 178 | What do hotel customers complain about? Text analysis using structural topic model (2019) | Customer dissatisfaction; Online hotel reviews; Structural topic model; Text mining; Trip advisor |

## 4. Bibliometric and Network Analysis

We briefly conducted a two-part data analysis comprising bibliometric analysis and network analysis (which included citation analysis and keyword network analysis) using NetMiner 4.0, a network analysis software that enables the analysis of not only network data but also unstructured text data. Using the bibliometric approach, we analyzed the frequency of titles and keywords in paper texts and abstracts to reveal a given paper's importance and then identify critical issues and trends. Unlike bibliometric analysis, the network analysis was performed to investigate core research issues and topics by constructing networks based on the co-citation of papers and co-occurrence of keywords.

### 4.1. Bibliometric Analysis

To perform the bibliometric analysis, we used additional data, such as the frequency of titles, authors, journals, and keywords. NetMiner was used to extract frequent words in the titles and keywords in selected papers and analyze the constructed networks.

The results of the bibliometric analysis are summarized in Table 3. Initially, 539 words were extracted from titles and 1595 words from abstracts. We determined the importance of these words based on their frequency of appearance. Words identified as important, shown in Table 3, include "service quality," "service," "hotel," "model," and "satisfaction." These results indicate that the included tourism service quality studies are mainly focused on the hotel industry and service satisfaction.

**Table 3.** The most frequently used words.

| No. | The Most Frequently Used Words in Titles | | No. | The Most Frequently Used Words in Abstracts | |
|---|---|---|---|---|---|
| | **Word** | **Frequency** | | **Word** | **Frequency** |
| 1 | service quality | 42 | 1 | service quality | 259 |
| 2 | customer service | 21 | 2 | study | 239 |
| | | | 3 | service | 154 |
| 4 | effect | 20 | 4 | customer | 145 |
| 5 | role | 19 | 5 | result | 132 |
| 6 | restaurant | 18 | 6 | hotel | 116 |
| 7 | hotel | 16 | 7 | research | 108 |
| 8 | perception | 11 | 8 | relationship | 106 |
| 9 | model satisfaction | 9 | 9 | satisfaction | 97 |
| | | | 10 | model | 95 |

*4.2. Network-Based Analysis: Citation Network Analysis*

Citation analysis has become more widespread because of its ability to objectively identify influential papers in a given area [16–18]. According to Ding and Cronin [12], citation analysis is primarily focused on identifying the popularity and significance of a published paper by counting how frequently that paper is cited by other papers [19].

To continue with our systematic literature review, we constructed a local citation network for analysis that determined how many times a published paper had been cited by other published papers within a local network comprised of the 178 initially selected papers. Then, we examined three structural characteristics of the citation network: density, distance, and clustering. Density measures the proportion of actual connections in a network relative to the number of connections possible, thereby revealing the size of the network. A large network is generally sparser. Distance refers to the average number of steps along the shortest paths for all pairs of published papers, indicating the degree of information efficiency on a network. Finally, the clustering coefficient reflects the degree of connection between a published paper and neighboring papers. This coefficient is based on the ratio of the number of existing links to the number of possible links among neighboring published papers. As shown in Table 4, the local citation network of published papers on service quality in tourism is relatively sparse and highly clustered.

**Table 4.** The characteristics of network structure.

| | **Papers** | **Density** | **Distance** | **Clustering Coefficient** |
|---|---|---|---|---|
| All | 178 | 0.01 | 1.397 | 1.614 |

We used the PageRank measure to further identify core papers within the subset of 178 papers. PageRank, introduced by Brin and Page [20], was created to prioritize web pages in the Google search engine [19]. PageRank can be used to measure "prestige," an important indicator of webpage quality, by revealing the number of times a paper is cited by other highly cited papers. The PageRank of paper A (denoted by PR(A)) in a network constructed with N papers can be computed as follows:

$$\frac{(1-d)}{N} + d\left(\frac{PR(T_1)}{C(T_1)} + \cdots\cdots + \frac{PR(T_n)}{C(T_n)}\right)$$

where paper $T_i$ has citations $C(T_i)$ and $d$ is a damping factor representing the fraction of random walks that continue to propagate along the citations. In our study, the parameter $d$ was chosen to be 0.85 based on Brin and Page [20].

Table 5 indicates the top 10 papers as analyzed by PageRank. These papers can be regarded as core papers in the field of tourism service quality and mainly address the importance of service quality

for customer satisfaction and its relationship with customer behavior. For example, Kim et al. [21] investigated the impact of tourism service quality on customer behavior according to customer satisfaction. Chen et al. [22] examined the determinants of customer participation in service encounters and their impact on customer loyalty. Ladhari et al. [23] analyzed the importance of service quality as a factor in dining satisfaction with regard to restaurant services. Ha and Jang [24] also studied the relationship between service quality and customer satisfaction regarding its effect on loyalty in Korean restaurants. Hutchinson et al. [25] attempted to clarify the impacts of service quality and customer satisfaction on customer behavioral intention. Additionally, some of the 10 papers dealt with service quality evaluation. For instance, Hsieh et al. [26] devised a service quality evaluation framework for hot spring hotels, and Martínez Caro and Martínez García [27] researched a comprehensive model to measure service quality in tourism-related sectors.

**Table 5.** Top 10 papers according to the PageRank algorithm.

| No. | Title | PageRank |
|---|---|---|
| | **Influential Papers** | |
| 1 | [21]<br>Influence of institutional DINESERV on customer satisfaction, return intention, and word-of-mouth | 0.018510 |
| 2 | [22]<br>Antecedents of mandatory customer participation in service encounters: An empirical study | 0.007353 |
| 3 | [28]<br>Construction and validation of the customer participation scale | |
| 4 | [23]<br>Determinants of dining satisfaction and post-dining behavioral intentions | 0.006856 |
| 5 | [24]<br>Effects of service quality and food quality: The moderating role of atmospherics in an ethnic restaurant segment | 0.005837 |
| 6 | [29]<br>Customer value in the hotel industry: What managers believe they deliver and what customer experience | 0.004211 |
| 7 | [26]<br>A service quality measurement architecture for hot spring hotels in Taiwan | 0.003915 |
| 8 | [30]<br>Using a modified grey relation method for improving airline service quality | 0.003915 |
| 9 | [25]<br>Understanding the relationships of quality, value, equity, satisfaction, and behavioral intentions among golf travelers | 0.003845 |
| 10 | [27]<br>Developing a multidimensional and hierarchical service quality model for the travel agency industry | 0.003835 |

Note: This selection was taken from the 178 papers that were initially selected.

Likewise, we further conducted a co-citation analysis to identify prevalent topics through the co-occurrence of two given papers in other papers [31]. The papers comprising the co-citation network were classified into several clusters, in which the links between the articles in the given cluster were greater than those of other clusters [31–33]. In our study, clusters' index Q was calculated by using the algorism [32] as follows:

$$Q = \frac{1}{2m} \sum_{vw} \left[ A_{vw} - \frac{k_v k_w}{2m} \right] \delta(c_v, c_w)$$

where $A_{vw}$ represents the weight of the edge between nodes $v$ and $w$, $k_v$ expresses the sum of the weights of the edges attached to node $v$ ($k_v = \sum_w A_{vw}$), $c_v$ is the community to which node $v$ is assigned, $\delta(i, j)$ is equal to 1 if $i = j$ and 0 otherwise, and $m = \frac{1}{2} \sum_{vw} A_{vw}$.

Table 6 shows the local citation network's division into seven topical issue clusters. To identify the core topical issues for each cluster, we determined the lead papers in each cluster by using the PageRank algorithm. Then, a general description for each cluster's topic was ascertained using these lead papers.

**Table 6.** The lead papers using the PageRank algorithm for each cluster.

| Clusters | | Top Three Lead Papers According to the PageRank Algorithm |
|---|---|---|
| 1 | [34] | A systematic approach to scale development in tourist shopping satisfaction: Linking destination attributes and shopping experience |
| | [35] | Enhancing service loyalty: The roles of delight, satisfaction, and service quality |
| | [36] | Festival attributes and perceptions: A meta-analysis of relationships with satisfaction and loyalty |
| 2 | [37] | An investigation of green hotel customers' decision formation: Developing an extended model of the theory of planned behavior |
| | [13] | Less is more: A new insight for measuring service quality of green hotels |
| | [38] | The impact of green practices in coastal tourism: An empirical investigation on an eco-labeled beach club |
| 3 | [29] | Customer value in the hotel industry: What managers believe they deliver and what customer experience |
| | [25] | Understanding the relationships of quality, value, equity, satisfaction, and behavioral intentions among golf travelers |
| | [39] | Pay for performance and work attitudes: The mediating role of employee-organization service value congruence |
| 4 | [40] | Influence of the quality of food, service, and physical environment on customer satisfaction and behavioral intention in quick-casual restaurants: Moderating role of perceived price |
| | [41] | Restaurant consumers repeat patronage: A service quality concern |
| | [42] | Foreign travelers' satisfaction with traditional Korean restaurants |
| 5 | [43] | Relationships among experiential marketing, experiential value, and customer satisfaction |
| | [44] | The role and effect of job satisfaction and empowerment on customers' perception of service quality: A study in the restaurant industry |
| | [45] | An exploratory examination of supervisor undermining, employee involvement climate, and the effects on customer perceptions of service quality in quick-service restaurants |
| 6 | [26] | A service quality measurement architecture for hot spring hotels in Taiwan |
| | [27] | Developing a multidimensional and hierarchical service quality model for the travel agency industry |
| | [46] | An examination of electronic video clips in the context of hotel Websites |
| 7 | [21] | Influence of institutional DINESERV on customer satisfaction, return intention, and word-of-mouth |
| | [22] | Determinants of dining satisfaction and post-dining behavioral intentions |
| | [24] | Effects of service quality and food quality: The moderating role of atmospherics in an ethnic restaurant segment |

Cluster 1 corresponded to tourist satisfaction from various perspectives. For example, Wong and Wan [34] explored tourists' shopping satisfaction and examined its dimensionality. Ahrholdt et al. [35] also investigated tourists' satisfaction and loyalty according to prior experience. Tanford and Jung [36] evaluated the factors contributing to tourists' festival satisfaction.

Cluster 2 related to sustainable issues in tourism. Han and Kim [37] examined tourists' intention to revisit a green hotel. Lee and Cheng [13] investigated tourists' decision-making process in terms of staying at a green hotel. Most recently, Merli et al. [38] addressed tourists' perception of green practices and the impact of these practices on their satisfaction and loyalty.

Cluster 3 corresponded to the value customers place on tourism service quality. For example, Nasution and Mavondo [29] viewed customer value from two perspectives: that of the service provider, and that of the customer. Hutchinson et al. [25] addressed the effect of service evaluations, including value, on customer intentions. Chiang and Birtch [39] researched the effect of service value congruence between the employee and organization on pay-for-performance and work attitudes.

Cluster 4 mainly focused on service quality in restaurants and its impact on customer satisfaction. For example, Ryu and Han [40] examined the effect of restaurant food and service quality on customer intention. Barber et al. [41] aimed to determine the relationship between restaurant cleanliness and customers' repeated patronage. Nam and Lee [42] investigated the factors related to foreign tourists' satisfaction with traditional Korean restaurants.

Cluster 5 generally included customer perceptions regarding service quality. For example, Yuan and Wu [43] focused on the emotional and functional values made by service quality. Gazzoli et al. [44] discussed the relationship of empowerment and job satisfaction to customers' perception of service quality. Mathe and Slevitch [45] explored the factors impacting customers' perception of service quality.

Cluster 6 contained various assessments of service quality to identify service quality's effect on customer satisfaction. For example, Hsieh et al. [26] analyzed customers' expectations for hotel service quality according to its five dimensions. Likewise, Martínez Caro and Martínez García [27] developed a comprehensive model to measure service quality in tourism. Kim and Mattila [46] also studied customer evaluations regarding hotel service through six distinct dimensions.

Lastly, Cluster 7 corresponded to how service quality affects customer behavior. For instance, Kim et al. [21] analyzed the effect of service quality on customers' intention to return and word-of-mouth endorsement. Ladhari et al. [23] determined dining satisfaction factors in terms of restaurant service and its relationship to customer behaviors such as loyalty and willingness to pay more. Similarly, Ha and Jang [24] examined the relationship between perceived quality by customers and restaurant loyalty.

To understand the evolution of tourism service quality research over time, we also conducted a dynamic co-citation analysis of analyzed articles to indicate the evolution/development of each cluster over time. (Table 7)

**Table 7.** Research focus of each cluster.

| Cluster | Main Topic |
| --- | --- |
| 1 | Tourist satisfaction |
| 2 | Sustainable issues in tourism |
| 3 | Value of service quality for customers |
| 4 | Restaurant service quality |
| 5 | Customer perception of tourism service quality |
| 6 | Tourism service quality assessment |
| 7 | Customer behavior |

As seen in Table 8, earlier publications corresponded to Clusters 1, 3, 5, 6, and 7. Significantly, development of the topics corresponding to Clusters 1, 5, and 7 has diminished while that of Clusters 2 and 4 has continued to grow. Furthermore, the number of research papers focusing on the topics in Clusters 2 and 3 has steadily increased.

**Table 8.** The number of published papers in each cluster (2008–2019).

| Year | Cluster 1 | Cluster 2 | Cluster 3 | Cluster 4 | Cluster 5 | Cluster 6 | Cluster 7 |
|------|-----------|-----------|-----------|-----------|-----------|-----------|-----------|
| 2008 | 1 | | 1 | | 1 | 2 | 1 |
| 2009 | | | 2 | | | | 1 |
| 2010 | | 1 | 1 | 1 | 1 | 2 | 3 |
| 2011 | | | 2 | 3 | | 2 | 1 |
| 2012 | | | | | | | 3 |
| 2013 | 1 | 1 | 1 | 1 | 1 | | 2 |
| 2014 | | | | 1 | 1 | 1 | 3 |
| 2015 | | | 2 | 1 | | | 3 |
| 2016 | | | 1 | | | 2 | 1 |
| 2017 | 2 | | 1 | 3 | 1 | 1 | 2 |
| 2018 | | 1 | 1 | 1 | | 2 | 1 |
| 2019 | | 1 | | 1 | | | |
| Total | 4 | 4 | 12 | 12 | 5 | 12 | 21 |

*4.3. Network-Based Analysis: Keyword Network Analysis*

After performing the citation network analysis, we executed a keyword network analysis based on 608 keywords extracted from 178 papers.

To conduct the keyword network analysis, we followed the process summarized in Table 9. First, we (1) constructed a keyword network using keywords extracted from premiere international tourism-focused academic journals (namely, *The Annals of Tourism Research, International Journal of Hospitality Management, Journal of Hospitality and Tourism Research, Journal of Travel Research*, and *Tourism Management*). We then (2) investigated the issues and topics related to service in the tourism sector utilizing network analysis. Finally, we (3) observed the changes in the issues and topics that occurred from 2008 to 2019.

**Table 9.** Keyword network analysis process.

| Analysis Process | | |
|---|---|---|
| Network construction | Keyword network construction | Refinement of search keywords |
| | | Construction of keyword network based on the frequency of keyword co-occurrence |
| | | Construction of commonly addressed keyword network based on component analysis |
| Network analysis | Keyword network analysis | Network centrality analysis (degree, betweenness, closeness) |
| | | Cluster analysis |
| | | Network centrality analysis by year |

More specifically, in the afore-mentioned step 1, we used 608 keywords to form a network for tourism service quality. Before constructing a keyword network, we refined the keywords extracted from 178 papers by standardizing keywords that had the same fundamental meaning. The rules used to refine the keywords [11] are as follows:

- Standardization into a singular form
- Removing redundant keywords
- Removing hyphens
- Avoidance of abbreviations
- Unification of synonyms
- Separation of multiple terms in a single keyword

Thus, standardization resulted in 604 relevant keywords from the original list of 629 keywords. To construct the network consisting of the most important, commonly referenced keywords, we then

performed component analysis using NetMiner. A component in a network indicates a group of nodes (papers) that are all connected to each other, representing commonly addressed issues and topics in the network.

### 4.3.1. Keyword Network Analysis: Network Centrality Analysis

After performing the component analysis, we were able to observe differences in the classification of the top 10 keywords across three different measures: degree, betweenness, and closeness centrality. As shown in Table 10, the differences in the top-ranked keywords according to the three measures implies that research on tourism service focuses on both broad and specific issues. The top 10 keywords according to the centrality measures are important in terms of their structural positions in the keyword network.

**Table 10.** Top 10 keywords across measures.

| Rank | Degree Centrality | Betweenness Centrality | Closeness Centrality |
|---|---|---|---|
| 1 | satisfaction | satisfaction | satisfaction |
| 2 | customer satisfaction | customer satisfaction | customer satisfaction |
| 3 | value | China | value |
| 4 | behavioral intention | behavioral intention | behavioral intention |
| 5 | online review | emotion | online review |
| 6 | equity | online review | equity |
| 7 | loyalty | service failure | loyalty |
| 8 | emotion | equity | emotion |
| 9 | perceived value | electronic word of mouth | perceived value |
| 10 | customer loyalty | perceived quality | customer loyalty |

The top 10 keywords according to the degree of centrality are "satisfaction," "customer satisfaction," "value," "behavioral intention," "online review," "equity," "loyalty," "emotion," "perceived value," and "customer satisfaction," and these words represent important keywords in terms of their structural position in the keywords network. These keywords have many connections with other keywords, which indicates that they represent major research issues in the field of tourism service quality.

The top 10 keywords according to betweenness centrality—"satisfaction," "customer satisfaction," "China," "behavioral intention," "emotion," "online review," "service failure," "equity," "electronic word-of-mouth," and "perceived quality"—play an important role in bridging separated groups of research themes. In other words, these keywords lie between two distinctive research themes.

Finally, the top 10 keywords in terms of closeness centrality are "satisfaction," "customer satisfaction", "value," "behavioral intention," "online review," "equity," "loyalty," "emotion," "perceived value," and "customer loyalty." These words were used with nearly all other keywords and themes in the network because a keyword with high closeness centrality is located in the center of the keyword network.

### 4.3.2. Keyword Network Analysis: Changes in Important Keywords Over Time

To address the changes in the important keywords over time, we compared the important keywords from the first nine years (2008–2016) with those from the three most recent years (2017–2019). We then compared high-ranked keywords across the three network measurements (see Table 11). It is important to note that the connections between keywords have accumulated over years, making it inherently difficult to investigate the evolution of keyword networks. In other words, although the keyword network constructed for a certain period of time offers information about the associations among the keywords for the published papers in that specific period, it is possible to exclude significant information regarding keyword associations in other periods [11]. The associations among keywords across different time periods do affect one another; thus, they are correlated [11]. This is a common issue when investigating the evolution of citation, author, and keyword networks. Comparing the keyword

network corresponding to data obtained from much earlier studies with that from more recent studies can mitigate the potential loss of information concerning recent changes in impactful keywords [11].

**Table 11.** Comparison of the top five keywords across three network measurements.

| | 2008–2016 | 2017–2019 | 2008–2016 | 2017–2019 | 2008–2016 | 2017–2019 |
|---|---|---|---|---|---|---|
| **Rank** | **Degree Centrality** | | **Closeness Centrality** | | **Betweenness Centrality** | |
| 1 | service quality | service quality | service quality | service quality | service quality | service quality |
| 2 | satisfaction | hospitality industry | satisfaction | trip advisor | satisfaction | customer satisfaction |
| 3 | customer satisfaction | trip advisor | customer satisfaction | loyalty | China | trip advisor |
| 4 | loyalty | loyalty | loyalty | hospitality industry | customer satisfaction | hospitality industry |
| 5 | equity | satisfaction | equity | satisfaction | emotion | loyalty |

Our comparison reveals some notable findings. "Service quality," "satisfaction," "customer satisfaction," and "loyalty" were the most important keywords for all three measures for both clustered time periods. Significantly, customer satisfaction-related keywords (such as "customer satisfaction," "satisfaction," and "loyalty") have received growing attention over the nine-year time period. Additionally, tourism management-related keywords (such as "trip advisor" and "hospitality industry") have become substantially more prevalent over the years.

## 5. Conclusions

Service quality has been established as an important economic impetus of tourism. To explore how this factor has been represented in past tourism literature, we conducted a systematic literature review combining bibliometric, citation network, and keyword network analysis. Furthermore, our study attempted to identify how important keywords have changed over time to capture the emerging critical issues and topical trends in tourism service research.

This study has significant implications for both theory and practice in several ways. First, due to the existence of diverse tourism sub-sectors across business settings, previous reviews on service quality have mainly focused on service quality with regard to specific themes pre-identified by the author. In contrast, the present study represents a more comprehensive literature review on service quality in tourism research by applying a systematic approach.

Second, we identified that the keyword network of service quality in tourism is relatively small, characterized by low density, short distance, and fewer clusters. It is possible that new issues and concepts related to service quality in tourism have not emerged as rapidly [11] because relatively large networks with less connectivity to neighboring research areas do not easily facilitate the creation of new issues and concepts.

Third, based on the citation analysis of research on tourism service quality, we identified critical issues most commonly discussed by influential papers. To conduct content analysis, we rigorously identified papers as "more important" by using not only the frequency of citations but also the degree of prestige, established through the PageRank measure. This is a significant methodological development, as previous studies have only used the frequency of citation to assess importance. The critical issues identified in this study concern how service quality affects tourists' behavior and service quality evaluation. Current research on service quality in tourism is still focused on the impact of service quality on customer satisfaction and behavior.

Fourth, this study extends and supports the previous systematic literature review on service quality provided by Lai et al. [15]. Their review suggested 17 research topics that comprise two main research streams: (1) service quality scares, and (2) the consequences of service quality. According to the co-citation analysis presented in this article, research on tourism service quality can actually be classified into seven topic clusters: (1) tourist satisfaction, (2) sustainable issues in tourism, (3) value of service quality for customers, (4) restaurant service quality, (5) customer perception of tourism service quality, (6) tourism service quality evaluation, and (7) service quality's influence on customer behavior.

Fifth, because important issues and topics were selected from a local citation network comprising 136 cited papers within the total 178 papers used in our study, it was difficult to identify comprehensive

issues and topics due to the exclusion of the remaining 42 papers from the network configuration. Therefore, we implemented a keyword network analysis to complement the drawbacks of citation and co-citation analysis. The keyword network analysis revealed differences in the classification of important keywords across centrality measures. Our results reveal some notable findings. "Satisfaction" and "customer satisfaction" are represented as major research issues in the area of service quality in tourism. Also, "satisfaction" and "customer satisfaction" have not only a high degree of centrality but also high betweenness and closeness centrality. These results indicate that these are important service quality issues, and the existing literature has focused on them. Thus, service quality issues related to "satisfaction" and/or "customer satisfaction" might be a good starting point for researching the overall topic of service quality in tourism.

Finally, to identify changes in and the development of topics over time, we performed a dynamic co-citation analysis. These results showed that "sustainable issues in tourism" and "restaurant service quality" have gained researchers' attention in recent years, whereas the focus on topics such as "tourist satisfaction," "customer perception of tourism service quality," and "service quality's influence on customer behavior" has decreased. This current trend suggests an increasing interest in investigating service quality regarding sustainability (green hotel and green practice) and restaurants. Thus, this paper suggests two research areas that deserve further investigation and research: (1) service quality and sustainability, and (2) service quality and restaurants.

Customer satisfaction-related keywords (such as "customer satisfaction," "satisfaction," and "loyalty") have received growing attention over the nine-year period used in the study, while the importance of tourism management-related keywords (such as "trip advisor" and "hospitality industry") has also substantially increased during the same period. The results of this study reveal the most frequently used words in research titles and abstracts in the field of tourism service quality. After using "service quality" as the main keyword to extract significant papers, "service quality" emerged as the most frequently used word in titles and abstracts. Moreover, "hotel" and "restaurant" are also included in the list of popular keywords. This indicates that tourism service quality is largely associated with hotels and restaurants compared to other tourist destinations.

Although our study has interesting implications for service quality tourism research, this study is not free of limitations. Despite providing a comprehensive systematic review of this area, the manual search method used to retrieve the articles may have excluded or overlooked other relevant articles. As the papers were retrieved only from SCOPUS, any related articles that could not be listed in one of these databases were excluded. In addition, we choose our keywords according to our research topic. The keywords used may not be exhaustive. Expanding the keywords to reflect "service quality" could result in a more exhaustive review of the field.

**Author Contributions:** J.P. and E.J. conceived of and designed the methodology; J.P. conducted the literature review and collected data; E.J. performed the analysis; J.P. and E.J. contributed to the interpretation of results and discussion; J.P. and E.J. wrote the paper.

**Funding:** This research received no external funding.

**Acknowledgments:** This research is (partially) supported by the BK21PLUS Research Fund for Korea University Business School.

**Conflicts of Interest:** The authors declare no conflict of interest.

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
