# Peer review of "Service Quality in Tourism: A Systematic Literature Review and Keyword Network Analysis"

_sustainability, doi:10.3390/su11133665_

Round 1
Reviewer 1 Report
The work of the authors is very interesting. However, in my opinion it suffers from some inaccuracies.
The authors comment that few studies have used a comprehensive perspective to analyze the quality of service in tourism and this is not true. The quality of the service has been widely studied in tourism and the work is not original. How does your research add to the literature?
Regarding the literature review, it should be expanded since there are many papers that have analyzed similar aspects and are not cited in their study.
In addition, in your study it would be good to incorporate a section of 'Discussion' where the authors compare the review of the literature with the new findings from their study.
The conclusions should inform theoretical contributions, contributions for professionals and limitations of the study.
Good luck,
Reviewer 2 Report
This paper provides a very important and up-to-date overview of the tourism service quality discussions in the academic literature. The relevance of this topic to sustainability issues is demonstrated in the text. I recommend to consider this paper for publication after certain improvements. My recommendations are given below.
1) This paper is a review, and, thus, the REVIEW type of papers should be selected instead of ARTICLE.
2) Lines 40-42: a bit unclear – do journals utilize SERVQUAL? How, if so?
3) The last paragraph of Introduction is not necessary.
4) Although this will require voluminous technical work, I encourage the authors to cite ALL 178 papers in the section 3 that should be named Materials and Methods.
5) Some methodological notes occur here and there among the results. These notes should be gathered together and moved to Materials and Methods.
6) This paper needs a new section Discussion. Here, practical importance of the research outcomes, comparisons to results of some other, somewhat similar studies (look at the tourism-related papers based on bibliometric approach), far-going tendencies, the research limitations (this appear in Conclusions, but these should be in Discussion), etc. should be discussed on 1-2 pages. The authors should explain clearly the general importance of their study – what does this mean in general?
7) Why not to draw 2-3 figures for this paper? This would make it more attractive to the readers.
Reviewer 3 Report
The paper deals with a very interesting topic and with up-to-date methods. The reading is flowing and the structure is clear. Few concerns:
-Section 2 (Literature Review). Given the fact that the literature review is actually the result of your study, I would suggest to anticipate the section on methodology in order to contextualize how the research has been designed. In my opinion the text in the current section 2 is related to methodological issues as well, and as such I would rather go for including it in the methodological part.
-Section 3 (Research Methodology). I would suggest to let the readers know more about the digital tools you used, such as NetMiner and PageRank. More specifically, it could be useful to know more on its functions and pros/cons.
-Section 3 (Research Methodology). It is not clear how you derived the clusters. Although you refer to previous studies, the readers would like to understand how you managed in your study (which criteria, which assumptions, which statistical features).
Round 2
Reviewer 1 Report
I appreciate the effort you put into reviewing this paper.
This revised version of your manuscript is more clear and I am sure that the paper has improved substantially.
